# Dissipativity Analysis of Large-Scale Networked Systems

**Yuanfei Sun** [1], **Jirong Wang** [2,3], **Huabo Liu** [1,4]*

1   School of Automation, Qingdao University, Qingdao 266071, China
2   School of Mechanical and Electrical Engineering, Qingdao University, Qingdao 266071, China
3   Weihai Innovation Insitute, Qingdao University, Weihai 264200, China
4   Shandong Key Laboratory of Industrial Control Technology, Qingdao 266071, China
*  Correspondence: hbliu@qdu.edu.cn

**Abstract:** This paper investigates the dissipativity analysis of large-scale networked systems with linear time-invariant dynamics. The networked system is composed of a large number of subsystems whose connections are arbitrary, and each subsystem can have different dynamics. A sufficient and necessary condition for the strict dissipativity analysis of the networked system is derived, which takes advantage of the block-diagonal structure of the system parameter matrix and the sparseness characteristics of the subsystem interconnections. Then, a necessary condition and a sufficient condition that depend only on a single subsystem parameter are given separately. Numerical simulations illustrate that compared with the existing results, the conditions suggested in this paper have higher computational efficiency in the dissipative analysis of large-scale networked systems.

**Keywords:** dissipativity; large-scale system; linear matrix inequality; networked system; sparseness





## 1. Introduction

In recent years, the research of large-scale networked systems has attracted great attentions [1–4]. The system can be considered as composed of a large number of subsystems with different spatial locations connected in a certain way [5]. Generally, subsystems exchange information with their neighbors straightforwardly and predictably, but the system often exhibits complicated dynamic behavior when seen as a whole. Such systems have extensive application background, including airplane formation flight [6], power network distributed system [7], automated highways [8], multi-agent formation systems [9], and so on. For such a complex system, the classic method of bringing all the subsystems together and analyzing it as a single large-scale system has very strict requirements on the calculation speed and accuracy of the computer, which will inevitably bring computational difficulties. Therefore, using the system structure to find more efficient computational conditions is of great engineering significance for the dissipativity analysis of large-scale networked systems.

There are already many results on the performance analysis of networked systems, but the research on dissipativity is not mature enough and needs to be further developed and improved. In 1972, the famous scholar Willems first put forward the concept of dissipativity [10,11]. Dissipativity describes the equilibrium relationship between the system's internal energy, which is a vital concept in theoretical research and practical application. Its essential meaning is that there is a non-negative energy function (called storage function) so that the energy supply rate of the system is always greater than the loss of energy inside the dynamic system. Based on Willems' work, many scholars have done in-depth studies on dissipativity and obtained rich results, which have played a significant role in the field of circuit, system, and control theory. Refs. [12,13] respectively studied the dissipative control problems of linear continuous-time and discrete systems based on linear matrix inequality (LMI) methods. A simplified mathematical model of the interconnected two-machine power system was established in [14], and its non-linear dynamic behavior

such as dissipativity properties was analyzed. In [15], the analysis and improvement of the dissipativity performance of interconnected passive systems are studied. For networked control systems, Ref. [16] obtained some new sufficient conditions by utilizing Lyapunov stability theory and LMI technology to ensure that the closed-loop system is finite-time limited and dissipative. In [17], a distributed controller was created to ensure the dissipativity of a networked system made up of dynamically coupled subsystems. Its control synthesis is done locally at the subsystem level and doesn't involve the relationship among subsystems, hence it has certain drawbacks. The linear dynamic system with the interconnection structure specified by the directed graph is studied in [18]. Based on the dissipativity inequality, an LMI for calculating system performance is established and the concept of local dissipativity is defined. Using the knowledge of graph theory to analyze large-scale networked systems has certain constraints on the dynamic characteristics and connection modes of subsystems, which has certain limitations in practical application [19,20].

Considering that the system structure of large-scale networked systems usually has sparse characteristics or specific structural forms [21–23], in the large-scale connected systems discussed in [24], the concept of internal input and output is introduced to represent the connections and functions among subsystems, and the connection relationship among sub-units of the entire system is described by subsystem connection matrix. This description method takes into account the situation where the dynamic characteristics of the subsystems are different and the connection relationship of the subsystems is arbitrary. The previously mentioned UAV formation flight refers to the arrangement of multiple UAVs in a certain formation so that they maintain in formation or change their relative positions within a limited scope during the flight. To maintain a certain formation shape, information interaction is required among the UAVs. In a centralized strategy, each UAV has to know information about the whole formation, demanding substantial information interaction. It is computationally intensive and requires the high performance of the airborne computer. In fact, each UAV can interact with its position, speed, attitude, and motion target with only the UAVs connected to it in the formation. In this way, the amount of computation is greatly reduced, and the system is relatively simple to implement. It is this sparse property or specific structural form among subsystem connections that we exploit to give more computationally efficient dissipativity criteria for large-scale networked systems. Dissipativity explains some of the energy losses and control problems of control systems and is a more general performance indicator of system performance.

In this paper, our objective is to reduce the computational burden of dissipativity analysis for large-scale network systems with a large number of subsystems. In general, the large-scale networked systems studied in this paper have the following characteristics. The first is that the scale of the system is large, including many subsystems. The second and most important point is that the interaction among subsystems of large-scale networked systems is usually sparse or has a specific structural form. We introduce intermediate variables in networked systems to describe the relationship among subsystems, which is more general and explicitly characterizes the structural characteristics of large-scale systems. In this regard, this paper proposes several new LMI conditions, which effectively use the block diagonal structure of the system parameter matrix and the sparsity of the subsystem connection matrix, avoiding the inverse computation of high-dimensional matrices. The simulation results indicate that the conditions proposed in this paper are more efficient than the existing results.

The following is the structure of the paper. The model and the definition of dissipativity for the networked system and some preliminary results are given in Section 2. In Section 3, some conditions for dissipative analysis of networked systems are given, and the relationship between these conditions and existing conditions is discussed. Some numerical simulation results are presented in Section 4. The research results of this paper are summarized in Section 5, and the direction of further research is also proposed here.

**Notation 1.** *The symbol R is used to denote the set of real numbers, and the vector space produced by real numbers of appropriate dimensions is denoted as $R^{\#}$. $col\{Z_i|_{i=1}^{L}\}$ denotes the vector/matrix stacked by $Z_i (i = 1, 2, \ldots, L)$, and $diag\{Z_i|_{i=1}^{L}\}$ signifies a block diagonal matrix with $Z_i$ as the i-th diagonal block. $\left\{Z_{ij}|_{i=1,j=1}^{i=M,j=N}\right\}$ stands a matrix with $M \times N$ blocks, and its i-th row j-th column block matrix being $Z_{ij}$. $0_n$ and $0_{n \times m}$ represent the n dimensional zero column vector and the $n \times m$ dimensional zero matrix respectively, the dimension subscript is omitted if there is no ambiguity, and the identity matrix I is the same. The superscript T represents the transpose of a matrix or vector, and $(*)^T WZ$ or $ZW(*)^T$ is shorthand for $Z^T WZ$ or $ZWZ^T$.*

## 2. System Description and Some Preliminaries

The networked system $\Gamma$ is consisted of $N$ linear time-invariant subsystems, and the $i$-th subsystem $\Gamma_i$ is defined by the state-space model below,

$$\begin{bmatrix} \dot{x}(t,i) \\ z(t,i) \\ y(t,i) \end{bmatrix} = \begin{bmatrix} A_{\text{TT}}(i) & A_{\text{TS}}(i) & B_{\text{T}}(i) \\ A_{\text{ST}}(i) & A_{\text{SS}}(i) & B_{\text{S}}(i) \\ C_{\text{T}}(i) & C_{\text{S}}(i) & D_{\text{T}}(i) \end{bmatrix} \begin{bmatrix} x(t,i) \\ v(t,i) \\ u(t,i) \end{bmatrix}, \tag{1}$$

in which $t$ and $i$ denote respectively for the temporal variable and the index number of a subsystem, $i = 1, 2, \ldots, N$. Moreover, $x(t,i)$ is the state vector of the $i$-th subsystem $\Gamma_i$ at time $t$. $y(t,i)$ and $u(t,i)$ represent the external output vector and external input vector of the $\Gamma_i$, respectively. $z(t,i)$ and $v(t,i)$ are the output vector to other subsystems and input vector from others, which is also called internal output vector and input vector. The connection relationship among subsystems can be expressed as

$$v(t) = \Phi z(t), \tag{2}$$

here, $v(t) = col\left\{v(t,i)|_{i=1}^{N}\right\}$ and $z(t) = col\left\{z(t,i)|_{i=1}^{N}\right\}$. $\Phi$ is called the subsystem connectivity matrix. We assume that each row of the matrix $\Phi$ has only one non-zero element equal to one and there are no columns in which all of the items are equal to zero. That means the internal output channels of a subsystem can affect the internal inputs channels of other subsystems, and some subsystem internal input channels depend on the internal output of multiple subsystems. This assumption, as explained in [23], does not jeopardize the generality of the adopted system model. Approximate power-law degree distribution widely exists in engineering systems, such as protein interaction networks, gene regulatory networks, power systems, the Internet, etc. [23]. In these systems, the dimension of the subsystem connection matrix $\Phi$ is usually much smaller than the state dimension of the system, and the interactions among subsystems are sparse.

In this paper, we assume that the dimensions of vectors $x(t,i)$, $v(t,i)$, $z(t,i)$, $u(t,i)$ and $y(t,i)$ are $m_{xi}$, $m_{vi}$, $m_{zi}$, $m_{ui}$ and $m_{yi}$, respectively. Based on the above assumptions and Equation (2), the dimension of the matrix $\Phi$ is $\sum_{i=1}^{N} m_{vi} \times \sum_{i=1}^{N} m_{zi}$. Then we can get $\Phi^T \Phi = \Sigma^2$ ,where $\Sigma^2 = diag\left\{\Sigma_j^2|_{j=1}^{N}\right\}$, $\Sigma_j^2 = diag\left\{m(i)|_{i=M_{z,j-1}+1}^{M_{z,j}}\right\}$, $M_{z,i} = \sum_{k=1}^{i} m_{zk}$ , $m(i)$ indicates the number of subsystems directly affected by the $i$-th element of the vector $z(t)$, $i = 1, \cdots, \sum_{k=1}^{N} m_{vk}$, $j = 1, \cdots, N$.

To simplify the mathematical derivation, we define the following matrix, $A_{*\#} = diag\left\{A_{*\#}(i)|_{i=1}^{N}\right\}$, $B_* = diag\left\{B_*(i)|_{i=1}^{N}\right\}$, $C_* = diag\left\{C_*(i)|_{i=1}^{N}\right\}$ and $D_* = diag\left\{D_*(i)|_{i=1}^{N}\right\}$ in which $*, \# = \text{T, S}$. By exploiting the connection relationship among subsystems, the dynamic system $\Gamma$ may be expressed equivalently in the following state-space form,

$$\begin{bmatrix} \dot{x}(t) \\ y(t) \end{bmatrix} = \begin{bmatrix} A & B \\ C & D \end{bmatrix} \begin{bmatrix} x(t) \\ u(t) \end{bmatrix}, \tag{3}$$

where

$$A = A_{\text{TT}} + A_{\text{TS}}\Phi(I - A_{\text{SS}}\Phi)^{-1}A_{\text{ST}},$$

$$B = B_T + A_{TS}\Phi(I - A_{SS}\Phi)^{-1}B_S,$$

$$C = C_T + C_S\Phi(I - A_{SS}\Phi)^{-1}A_{ST},$$

$$D = D_T + C_S\Phi(I - A_{SS}\Phi)^{-1}B_S.$$

**Note:** Well-posedness is very important in system design, and ill-posed systems are usually difficult to control or impossible to estimate [25–27]. Therefore, this paper assumes that System Γ is well-posed, which means that $(I - A_{SS}\Phi)^{-1}$ exists.

This paper intends to establish computationally effective conditions for the dissipativity analysis of large-scale networked systems Γ. The concept of dissipativity is very important in the system, whether from the perspective of theoretical research or the perspective of practical application. Roughly speaking, dissipative systems can be described as such properties. At any time, the energy that the system may provide cannot exceed the energy already supplied. We first describe the definition of dissipativity for System Γ.

The definition is related to the supply function. For the *i*-th subsystem $\Gamma_i$, its supply function is defined as

$$s_i(u(t,i), y(t,i)) = \left[\begin{array}{c} y(t,i) \\ u(t,i) \end{array}\right]^T Q(i) \left[\begin{array}{c} y(t,i) \\ u(t,i) \end{array}\right], \tag{4}$$

where $Q(i)$ is a symmetric matrix of suitable dimensions.

**Definition 1.** *The large-scale networked system (1) and (2) with $x(0,i) = 0$ is said to be dissipative with supply function $s_i(u(t,i), y(t,i))$ if and only if there is a matrix $P(i) \geq 0$, such that,*

$$\int_{t_0}^{t_1} \sum_{i=1}^N s_i(u(t,i), y(t,i)) dt \geq \sum_{i=1}^N x^T(t_1, i)P(i)x(t_1, i) - \sum_{i=1}^N x^T(t_0, i)P(i)x(t_0, i) \tag{5}$$

*holds for all $t_0 \leq t_1$.*

According to the definition, the supply function can be interpreted as the energy transferred to the system, which means that within a period of time $[t_0, t_1]$, as long as $\int_{t_0}^{t_1} \sum_{i=1}^N s_i(u(t,i), y(t,i)) dt$ is positive, the system will work normally, otherwise, the system will not work. $\sum_{i=1}^N x^T(t_1, i)P(i)x(t_1, i) - \sum_{i=1}^N x^T(t_0, i)P(i)x(t_0, i)$ represents the actual energy consumption of the system after the time interval $t_1 - t_0$. Therefore, Equation (5) shows that in any time period $[t_0, t_1]$, the energy change inside the system will not exceed the energy supplied by the outside.

It can be seen from the following derivation that the definition of dissipativity for the networked system (1) and (2) are consistent with the one in [28] based on (3).

The supply function based on the large-scale networked system (1) and (2) is as follows,

$$s(u(t), y(t)) = \sum_{i=1}^N s_i(u(t,i), y(t,i)) = \left[\begin{array}{c} \left[\begin{array}{c} y(t,1) \\ u(t,1) \end{array}\right] \\ \left[\begin{array}{c} y(t,2) \\ u(t,2) \end{array}\right] \\ \vdots \\ \left[\begin{array}{c} y(t,N) \\ u(t,N) \end{array}\right] \end{array}\right]^T Q \left[\begin{array}{c} \left[\begin{array}{c} y(t,1) \\ u(t,1) \end{array}\right] \\ \left[\begin{array}{c} y(t,2) \\ u(t,2) \end{array}\right] \\ \vdots \\ \left[\begin{array}{c} y(t,N) \\ u(t,N) \end{array}\right] \end{array}\right]. \tag{6}$$

The supply function of System (3) is

$$s(u(t), y(t)) = \begin{bmatrix} y(t) \\ u(t) \end{bmatrix}^T Q_1 \begin{bmatrix} y(t) \\ u(t) \end{bmatrix}$$

$$= \begin{bmatrix} \begin{bmatrix} y(t,1) \\ y(t,2) \\ \vdots \\ y(t,N) \end{bmatrix} \\ \begin{bmatrix} u(t,1) \\ u(t,2) \\ \vdots \\ u(t,N) \end{bmatrix} \end{bmatrix}^T Q_1 \begin{bmatrix} \begin{bmatrix} y(t,1) \\ y(t,2) \\ \vdots \\ y(t,N) \end{bmatrix} \\ \begin{bmatrix} u(t,1) \\ u(t,2) \\ \vdots \\ u(t,N) \end{bmatrix} \end{bmatrix} = \begin{bmatrix} \begin{bmatrix} y(t,1) \\ u(t,1) \end{bmatrix} \\ \begin{bmatrix} y(t,2) \\ u(t,2) \end{bmatrix} \\ \vdots \\ \begin{bmatrix} y(t,N) \\ u(t,N) \end{bmatrix} \end{bmatrix}^T Q \begin{bmatrix} \begin{bmatrix} y(t,1) \\ u(t,1) \end{bmatrix} \\ \begin{bmatrix} y(t,2) \\ u(t,2) \end{bmatrix} \\ \vdots \\ \begin{bmatrix} y(t,N) \\ u(t,N) \end{bmatrix} \end{bmatrix}, \tag{7}$$

in which $y(t) = col\{y(t,i)|_{i=1}^N\}$, $u(t) = col\{u(t,i)|_{i=1}^N\}$, $Q = diag\{Q(i)|_{i=1}^N\}$ and

$$Q = \begin{bmatrix} I & 0 & 0 & 0 & & & 0 & 0 \\ 0 & \vdots & I & \vdots & & & \vdots & \vdots \\ \vdots & \vdots & 0 & \vdots & & & \vdots & \vdots \\ \vdots & \vdots & \vdots & \vdots & & 0 & \vdots \\ 0 & 0 & 0 & 0 & \cdots\cdots & I & 0 \\ 0 & I & 0 & 0 & & & 0 & 0 \\ \vdots & \vdots & \vdots & I & & & \vdots & \vdots \\ \vdots & \vdots & \vdots & \vdots & & & \vdots & \vdots \\ \vdots & \vdots & \vdots & \vdots & & \vdots & 0 \\ 0 & 0 & 0 & 0 & & & 0 & I \end{bmatrix} Q_1(*)^T.$$

For the convenience of the following discussion, we introduce the following preliminary results that need to be used.

**Lemma 1** ([29])**.** *For matrices L and U with compatible dimensions, there is a scalar $\alpha > 0$ such that,*

$$LU + U^T L^T \le \alpha LL^T + \alpha^{-1} U^T U. \tag{8}$$

**Lemma 2** ([30])**.** *Given symmetric matrices F and G with appropriate dimensions, if $v^T F v > 0$ can be obtained for every non-zero vector v satisfying $v^T G v = 0$, then there must be a real number r such that F+rG is positive definite, and vice versa.*

**Lemma 3** ([29])**.** *For an LMI in the form of an $M \times M(M \ge 1)$ block matrix: $G(P) < 0$, except for the symmetric independent variable matrix P, other known coefficient matrices or constant matrices are all block diagonal matrices of appropriate dimensions, and all have $N(N > 1)$ diagonals. If it is divided into blocks, there is a full block feasible solution P for this LMI, and there must be a feasible solution for the diagonal division of the appropriate dimension.*

## 3. Dissipativity Analysis

In [28], the dissipativity criterion of System (3) is proposed.

**Lemma 4.** *Assume that the networked system $\Gamma$ is controllable. Then, System (3) is strictly dissipative with the supply function $s(u(t), y(t))$ if and only if there exists a matrix $P > 0$ such that,*

$$\begin{bmatrix} A^T P + PA & PB \\ B^T P & 0 \end{bmatrix} - \begin{bmatrix} C & D \\ 0 & I \end{bmatrix}^T Q \begin{bmatrix} C & D \\ 0 & I \end{bmatrix} < 0. \tag{9}$$

Note that the matrices $A$, $B$, $C$ and $D$ in the condition of Lemma 4 all contain $(I - A_{SS}\Phi)^{-1}$ terms. Although the subsystem connection matrix $\Phi$ is sparse and the system

parameter $A_{*\#}$, $B_*$, $C_*$ and $D_*$ with $*, \# = $ T, S are block diagonal, the matrix $(I - A_{SS}\Phi)^{-1}$ is generally dense. When there are a large number of subsystems in large-scale networked systems, the calculation of matrices $A$, $B$, $C$, and $D$ involves the inversion of high-dimensional matrices. Therefore, when the scale of the networked system increases, the computational complexity of Equation (9) will become very high.

Lemma 4 is a dissipative analysis condition based on lumped networked model. Due to the establishment of the lumped model, the connection relationship among subsystems is hidden inside the parameters, and its structural information is not effectively utilized. As a result, for large-scale networked systems, the use of this condition for dissipative testing will inevitably bring computational difficulties and even cannot be calculated.

Then, to reduce the computational difficulty caused by the increase of system scale, we establish a computationally efficient sufficient, and necessary condition for the strict dissipativity analysis of large-scale networked systems. This condition effectively utilizes the sparse structure of the subsystem connection matrix $\Phi$ in the networked system, that is, each subsystem is only connected to a limited number of other subsystems.

**Theorem 1.** *Assume that the networked system $\Gamma$ is controllable. Then, System $\Gamma$ is strictly dissipative with the supply function $s_i(u(t,i),y(t,i))$ if and only if there exists a symmetric positive definite matrix P and a positive scalar h such that,*

$$
(*)^T \begin{bmatrix} \begin{bmatrix} 0 & P \\ P & 0 \end{bmatrix} & \\ & -Q \end{bmatrix} \begin{bmatrix} 0 & I & 0 \\ A_{TS} & A_{TT} & B_T \\ C_S & C_T & D_T \\ 0 & 0 & I \end{bmatrix}
$$
$$
-h \times (*)^T \begin{bmatrix} I & -\Phi \\ -\Phi^T & \Sigma^2 \end{bmatrix} \begin{bmatrix} I & \begin{bmatrix} 0 & 0 \end{bmatrix} \\ A_{SS} & \begin{bmatrix} A_{ST} & B_S \end{bmatrix} \end{bmatrix} < 0. \tag{10}
$$

**Proof of Theorem 1.** Equation (9) can be expressed equivalently as follows,

$$
(*)^T \begin{bmatrix} \begin{bmatrix} 0 & P \\ P & 0 \end{bmatrix} & \\ & -Q \end{bmatrix} \begin{bmatrix} I & 0 \\ A & B \\ C & D \\ 0 & I \end{bmatrix} < 0. \tag{11}
$$

We express Equation (11) in the following equivalent form,

$$
(*)^T \begin{bmatrix} \begin{bmatrix} 0 & P \\ P & 0 \end{bmatrix} & \\ & -Q \end{bmatrix} \begin{bmatrix} I & 0 & 0 & 0 \\ 0 & 0 & I & 0 \\ 0 & 0 & 0 & I \\ 0 & I & 0 & 0 \end{bmatrix} \begin{bmatrix} I & 0 \\ 0 & I \\ A & B \\ C & D \end{bmatrix} < 0. \tag{12}
$$

Matrices $A$, $B$, $C$, and $D$ can be written as follows,

$$
\begin{bmatrix} A & B \\ C & D \end{bmatrix} = \begin{bmatrix} A_{TT} & B_T \\ C_T & D_T \end{bmatrix} + \begin{bmatrix} A_{TS} \\ C_S \end{bmatrix} \Phi(I - A_{SS}\Phi)^{-1} \begin{bmatrix} A_{ST} & B_S \end{bmatrix}. \tag{13}
$$

Substituting the above formula into Equation (12), we can get that,

$$
(*)^T \begin{bmatrix} \begin{bmatrix} 0 & P \\ P & 0 \end{bmatrix} & \\ & -Q \end{bmatrix} \begin{bmatrix} 0 & I & 0 \\ A_{TS} & A_{TT} & B_T \\ C_S & C_T & D_T \\ 0 & 0 & I \end{bmatrix}
$$
$$
\times \begin{bmatrix} \Phi(I - A_{SS}\Phi)^{-1}A_{ST} & \Phi(I - A_{SS}\Phi)^{-1}B_S \\ I & 0 \\ 0 & I \end{bmatrix} < 0. \tag{14}
$$

Then we define matrices $F$, $M$, and $K$ as follows,

$$F = (*)^T \begin{bmatrix} -\begin{bmatrix} 0 & P \\ P & 0 \end{bmatrix} & \\ & Q \end{bmatrix} \begin{bmatrix} 0 & I & 0 \\ A_{\text{TS}} & A_{\text{TT}} & B_{\text{T}} \\ C_{\text{S}} & C_{\text{T}} & D_{\text{T}} \\ 0 & 0 & I \end{bmatrix}, \tag{15}$$

$$M = \begin{bmatrix} \Phi(I - A_{\text{SS}}\Phi)^{-1} \begin{bmatrix} A_{\text{ST}} & B_{\text{S}} \end{bmatrix} \\ \begin{bmatrix} I & 0 \\ 0 & I \end{bmatrix} \end{bmatrix}, \tag{16}$$

$$K = \begin{bmatrix} I & -\Phi \end{bmatrix} \begin{bmatrix} I & \begin{bmatrix} 0 & 0 \end{bmatrix} \\ A_{\text{SS}} & \begin{bmatrix} A_{\text{ST}} & B_{\textbf{S}} \end{bmatrix} \end{bmatrix}. \tag{17}$$

Obviously, $M^T(-F)M < 0$. When $v = M\zeta$, $\zeta \in R^{\#}$, for any $v \neq 0$, we can get that $Kv = 0$, which means $v^T F v > 0$. According to Lemma 2, there must be a real number $h$ such that,

$$(*)^T \begin{bmatrix} -\begin{bmatrix} 0 & P \\ P & 0 \end{bmatrix} & \\ & Q \end{bmatrix} \begin{bmatrix} 0 & I & 0 \\ A_{\text{TS}} & A_{\text{TT}} & B_{\text{T}} \\ C_{\text{S}} & C_{\text{T}} & D_{\text{T}} \\ 0 & 0 & I \end{bmatrix}$$
$$+ h \times (*)^T \begin{bmatrix} I & -\Phi \\ -\Phi^T & \Sigma^2 \end{bmatrix} \begin{bmatrix} I & \begin{bmatrix} 0 & 0 \end{bmatrix} \\ A_{\text{SS}} & \begin{bmatrix} A_{\text{ST}} & B_{\textbf{S}} \end{bmatrix} \end{bmatrix} > 0. \tag{18}$$

That is,

$$(*)^T \begin{bmatrix} \begin{bmatrix} 0 & P \\ P & 0 \end{bmatrix} & \\ & -Q \end{bmatrix} \begin{bmatrix} 0 & I & 0 \\ A_{\text{TS}} & A_{\text{TT}} & B_{\text{T}} \\ C_{\text{S}} & C_{\text{T}} & D_{\text{T}} \\ 0 & 0 & I \end{bmatrix}$$
$$- h \times (*)^T \begin{bmatrix} I & -\Phi \\ -\Phi^T & \Sigma^2 \end{bmatrix} \begin{bmatrix} I & \begin{bmatrix} 0 & 0 \end{bmatrix} \\ A_{\text{SS}} & \begin{bmatrix} A_{\text{ST}} & B_{\textbf{S}} \end{bmatrix} \end{bmatrix} < 0. \tag{19}$$

The characterization of the left term of Equation (19) shows that if the inequality has a solution, then there must be $h > 0$. So far, the necessity has been proved. Then, multiply the left and right sides of Equation (10) by the matrices $M$ and $M^T$ respectively, and direct algebraic operations can complete the sufficiency proof.

$$(*)^T \begin{bmatrix} \begin{bmatrix} 0 & P \\ P & 0 \end{bmatrix} & \\ & -Q \end{bmatrix} \begin{bmatrix} 0 & I & 0 \\ A_{\text{TS}} & A_{\text{TT}} & B_{\text{T}} \\ C_{\text{S}} & C_{\text{T}} & D_{\text{T}} \\ 0 & 0 & I \end{bmatrix} \begin{bmatrix} \Phi(I - A_{\text{SS}}\Phi)^{-1} \begin{bmatrix} A_{\text{ST}} & B_{\text{S}} \end{bmatrix} \\ \begin{bmatrix} I & 0 \\ 0 & I \end{bmatrix} \end{bmatrix}$$
$$- h \times (*)^T \begin{bmatrix} I & -\Phi \\ -\Phi^T & \Sigma^2 \end{bmatrix} \begin{bmatrix} I & \begin{bmatrix} 0 & 0 \end{bmatrix} \\ A_{\text{SS}} & \begin{bmatrix} A_{\text{ST}} & B_{\textbf{S}} \end{bmatrix} \end{bmatrix} \begin{bmatrix} \Phi(I - A_{\text{SS}}\Phi)^{-1} \begin{bmatrix} A_{\text{ST}} & B_{\text{S}} \end{bmatrix} \\ \begin{bmatrix} I & 0 \\ 0 & I \end{bmatrix} \end{bmatrix} < 0. \tag{20}$$

The proof is completed. □

It can be seen that the condition in Lemma 4 hides the connection relationship among subsystems inside the parameters, while the left side of Equation (10) in Theorem 1 linearly depends on the symmetric matrix $P$, and the structure of the system is specifically reflected in it, which can effectively make use of the sparse structure of the subsystem connection matrix. Furthermore, the matrices $A_{*\#}$, $B_*$, $C_*$ and $D_*$ with $*, \# = \text{T}, \text{S}$ are all block diagonal, and large-scale networked systems are sparse. Combined with the research on sparse semi-definite programming problems [31–33], when the system is relatively large, the computational complexity of solving the above sparse LMI is frequently lower than the condition in Lemma 4. This aspect can also be explained in subsequent numerical

simulations. It is worth noting that the condition of Theorem 1 does not bring conservatism compared with Lemma 4, a dissipative criterion based on the lumped description.

When there are a huge number of subsystems, the strict dissipativity analysis using the condition in Theorem 1 may still encounter computational difficulties. To overcome this difficulty, we further explore the structural characteristics of the subsystem connection matrix $\Phi$, and put forward the conditions for strictly dissipative analysis based on the parameters of each subsystem.

A simple derivation leads to the following relationship,

$$\begin{bmatrix} I & -\Phi \\ -\Phi^T & \Phi^T\Phi \end{bmatrix} \leq 2\left( \begin{bmatrix} I \\ 0 \end{bmatrix} \begin{bmatrix} I & 0 \end{bmatrix} + \begin{bmatrix} 0 \\ \Phi^T \end{bmatrix} \begin{bmatrix} 0 & \Phi \end{bmatrix} \right). \tag{21}$$

Combined with Lemma 4 and the properties of the subsystem connection matrix, on the basis of Equation (10), the necessary condition for the strict dissipativity analysis that only depends on the parameters of a single subsystem can be obtained.

**Theorem 2.** *Assume that the networked system $\Gamma$ is controllable. A necessary condition for the strict dissipativity of System $\Gamma$ with the supply function $s_i(u(t,i), y(t,i))$ is that each subsystem has a symmetric positive definite matrix $P(i)$ and a positive scalar $h$ such that,*

$$(*)^T \begin{bmatrix} \begin{bmatrix} 0 & P(i) \\ P(i) & 0 \end{bmatrix} & \\ & -Q(i) \end{bmatrix} \begin{bmatrix} 0 & I & 0 \\ A_{\mathrm{TS}}(i) & A_{\mathrm{TT}}(i) & B_{\mathrm{T}}(i) \\ C_{\mathrm{S}}(i) & C_{\mathrm{T}}(i) & D_{\mathrm{T}}(i) \\ 0 & 0 & I \end{bmatrix}$$

$$-h \times (*)^T \begin{bmatrix} I & \\ & \Sigma_i^2 \end{bmatrix} \begin{bmatrix} I & \begin{bmatrix} 0 & 0 \end{bmatrix} \\ A_{\mathrm{SS}}(i) & \begin{bmatrix} A_{\mathrm{ST}}(i) & B_{\mathrm{S}}(i) \end{bmatrix} \end{bmatrix} < 0. \tag{22}$$

For large-scale networked systems, sometimes the parameters of multiple subsystems are the same. In this case, using Theorem 2 is more efficient. A sufficient condition for strict dissipativity analysis which only depends on the parameters of a single subsystem is given below.

**Theorem 3.** *Assume that the networked system $\Gamma$ is controllable. Then, System $\Gamma$ is strictly dissipative with the supply function $s_i(u(t,i), y(t,i))$ if there exists a symmetric positive definite matrix $P(i)$ and real number $h_2 \geq h_1 \geq 0$ (or $h_1 \leq h_2 \leq 0$) for each subsystem such that,*

$$(*)^T \begin{bmatrix} \begin{bmatrix} 0 & P(i) \\ P(i) & 0 \end{bmatrix} & \\ & -Q(i) \end{bmatrix} \begin{bmatrix} 0 & I & 0 \\ A_{\mathrm{TS}}(i) & A_{\mathrm{TT}}(i) & B_{\mathrm{T}}(i) \\ C_{\mathrm{S}}(i) & C_{\mathrm{T}}(i) & D_{\mathrm{T}}(i) \\ 0 & 0 & I \end{bmatrix}$$

$$-(*)^T \begin{bmatrix} h_1 I & \\ & -h_2\Sigma_i^2 \end{bmatrix} \begin{bmatrix} I & \begin{bmatrix} 0 & 0 \end{bmatrix} \\ A_{\mathrm{SS}}(i) & \begin{bmatrix} A_{\mathrm{ST}}(i) & B_{\mathrm{S}}(i) \end{bmatrix} \end{bmatrix} < 0. \tag{23}$$

**Proof of Theorem 3.** From Lemma 1, we can get

$$\begin{bmatrix} I & -\Phi \\ -\Phi^T & \Phi^T\Phi \end{bmatrix} \geq (1-\alpha) \begin{bmatrix} I \\ 0 \end{bmatrix} (*)^T + \left(1 - \tfrac{1}{\alpha}\right) \begin{bmatrix} 0 \\ \Phi^T \end{bmatrix} (*)^T. \tag{24}$$

Using the above formula and the conclusion in Theorem 1, one can obtain a sufficient condition for System $\Gamma$ to be strictly dissipative is the existence of a symmetric positive definite matrix $P$ and two real numbers $h > 0$, $\alpha > 0$, such that,

$$(*)^T \left[ \begin{bmatrix} 0 & P \\ P & 0 \end{bmatrix} \quad -Q \right] \begin{bmatrix} 0 & I & 0 \\ A_{\mathrm{TS}} & A_{\mathrm{TT}} & B_{\mathrm{T}} \\ C_{\mathrm{S}} & C_{\mathrm{T}} & D_{\mathrm{T}} \\ 0 & 0 & I \end{bmatrix}$$

$$-h \times (*)^T \left( (1-\alpha) \begin{bmatrix} I \\ 0 \end{bmatrix} (*)^T + \left(1 - \frac{1}{\alpha}\right) \begin{bmatrix} 0 \\ \Phi^T \end{bmatrix} (*)^T \right) \begin{bmatrix} I & \\ A_{\mathrm{SS}} & \begin{bmatrix} 0 & 0 \\ A_{\mathrm{ST}} & B_{\mathrm{S}} \end{bmatrix} \end{bmatrix} < 0. \tag{25}$$

Let $h_1 = (1-\alpha)h$, $h_2 = -(1-\alpha^{-1})h$, we can get $h_2 = \alpha^{-1}h_1$. Therefore, when $\alpha \leq 1$, $h_2 \geq h_1 \geq 0$; when $\alpha \geq 1$, $h_1 \leq h_2 \leq 0$. The proof can be completed by combining Lemma 3. □

Compared with Theorem 1, the left side of Equation (23) in Theorem 3 is linearly related to the matrix $P(i)$, and its dimension is entirely governed by the dimension of the subsystem $\Gamma_i$. When the state dimension of each subsystem is fixed, the computational complexity of Equation (23) only linearly depends on the number of subsystems $N$. Therefore, Theorem 3 has a substantially higher computing efficiency than Theorem 1 for large-scale networked systems. However, it should be noted that Theorems 2 and 3 are conservative.

## 4. Numerical Simulations

Several numerical simulations are employed in this section to demonstrate the efficacy of the strict dissipativity conditions presented in this paper. The simulation experiments are performed on a laptop computer with an Intel(R) Core(TM) i5-3230M CPU @ 2.60 GHz 2.60 GHz and 6 G RAM. In these simulations, we assume that $m_{ui} = m_{xi} = m_{vi} = m_{zi} = m_{yi} = 2$. Furthermore, all the parameters of the subsystem are independent of each other, and the parameters of each subsystem are randomly generated according to a continuous uniform distribution with an interval of $[-0.9, 0.9]$. The subsystem connection matrix is randomly generated, but there is only one non-zero element 1 in each row and column.

The conditions in Lemma 4, Theorem 1, and Theorem 3 are used to verify the strict dissipativity of the system. Among them, the conditions in Lemma 4 and Theorem 3 are calculated by the LMI toolbox provided by MATLAB, and the condition in Theorem 1 is calculated by the sparse solvers DSDP. For System $\Gamma$ introduced in this paper, we generate 10 systems for calculation, and the average value and standard deviation of system dissipativity analysis calculation time are obtained. Tables 1 and 2 give some results when the number of subsystems is among 2 and 45.

**Table 1.** Average of calculation time.

| Subsystem Number | Lemma 4 (s) | Theorem 1 (s) | Theorem 3 (s) |
|---|---|---|---|
| 2 | 0.256809 | 0.113630 | 0.280457 |
| 10 | 0.506572 | 0.357334 | 0.332980 |
| 20 | 4.718364 | 1.825162 | 0.550354 |
| 30 | 49.843413 | 10.102070 | 1.132749 |
| 38 | 160.804551 | 25.194260 | 1.990903 |
| 40 | 201.484656 | 31.942283 | 2.235822 |

**Table 2.** Standard deviation of calculation time.

| Subsystem Number | Lemma 4 (s) | Theorem 1 (s) | Theorem 3 (s) |
|---|---|---|---|
| 2 | 0.005145 | 0.019887 | 0.003355 |
| 10 | 0.017458 | 0.038721 | 0.005182 |
| 20 | 0.035518 | 0.051297 | 0.005435 |
| 30 | 0.585151 | 0.160743 | 0.011004 |
| 38 | 1.305206 | 0.317135 | 0.017758 |
| 40 | 1.806196 | 0.389699 | 0.092795 |

The tables show that the calculation time of the above three methods all increases with the increase of the number of subsystems. When the number of subsystems is 10 or less, the computational efficiency based on Lemma 4 is comparable to that of Theorem 1 and Theorem 3. This is because the dimensionality of the matrix inequality in Theorem 1 is higher than that in Lemma 4, and Theorem 3 requires several inequalities to be verified. With the expansion of the number of subsystems, when the number of subsystems is 20, 30, 40, 45, the ratio of calculation time based on the conditions in Lemma 4 and Theorem 1 is 1.0957, 1.3414, 1.3911, 1.4581. The ratio of the average computation time becomes larger and larger, which means that the computational efficiency of Theorem 1 is improved to some extent. Because the condition in Lemma 4 requires operations such as inversion of high-dimensional matrices. Clearly, Theorem 3 is more computationally efficient than both Lemma 4 and Theorem 1. This is due to the fact that the conditions of Theorem 3 are tested based on individual subsystem parameters, and their computational complexity only increases linearly with the number of subsystems N. In addition, due to the limitation of computer memory, the conditions in Lemma 4 and Theorem 1 may not be calculated, but Theorem 3, which is tested independently for each subsystem, can still be calculated. Therefore, Theorem 3 has more computational advantages in the dissipativity analysis of large-scale networked systems. It should be noted that Theorem 3 is conservative compared to Lemma 4 and Theorem 1.

## 5. Conclusions

This paper investigates the strict dissipativity of networked systems composed of a large number of subsystems. At first, according to the model of large-scale networked systems, the definition of the dissipativity of networked systems is given in this paper. Then, we study the dissipative criteria of networked systems. For large-scale networked systems, when the number of subsystems is large, the performance analysis using the existing linear system theory will encounter computational difficulties. Some LMI-form conditions for dissipativity analysis of large-scale networked systems are derived. Among them, Theorem 1 is a necessary and sufficient condition, which effectively utilizes the block diagonal structure of the system parameter matrix and the sparsity of the subsystem connection matrix. Combined with the use of sparse semidefinite programming tools, it is more efficient than the lumped analysis method for medium-scale networked systems. In addition, the proposed sufficient condition, and necessary condition only depend on the parameters of a single subsystem, which are more suitable for the dissipative analysis of networked systems with a large number of subsystems, but they are conservative compared with other conditions.

Regarding large-scale networked systems, the design of distributed controllers to ensure the dissipativity of large-scale networked systems will be investigated in further research. For instance, in UAV formation flight, relying on a centralized controller to observe the entire formation and control all UAVs at once is both impractical and increases operational costs in engineering applications. A more reasonable option would be to decentralize the controller to each UAV platform and achieve the overall objective by interacting and sharing information between platforms. Therefore, if the structural information of the network topology is capable of being fully utilized and a distributed control strategy that relies on local information sharing is adopted, the amount of data transmission in the network will be greatly reduced and the computational efficiency will be raised. Furthermore, the presence of quantization errors, time delays, data packet loss, and other phenomena when communicating networked among subsystems or among subsystems and their local controllers will be explored. In practical engineering applications, network connections would possibly be non-idealized, and the arrival of information delivered is frequently unable to be achieved immediately.

**Author Contributions:** Conceptualization, Y.S. and H.L.; methodology, Y.S.; software, Y.S.; validation, H.L.; formal analysis, Y.S.; investigation, Y.S.; resources, J.W. and H.L.; data curation, J.W.; writing—original draft preparation,Y.S.; writing—review and editing, Y.S.; visuConceptualizationalization, J.W. and H.L.; supervision, J.W. and H.L.; project administration, J.W. and H.L.; funding acquisition, J.W. and H.L. All authors have read and agreed to the published version of the manuscript.

**Funding:** This work is supported by the National Natural Science Foundation of China under grant number 62273189; by the Shandong Provincial Natural Science Foundation under grant number ZR2019MF063 and ZR2020MF064.

**Institutional Review Board Statement:** Not applicable.

**Informed Consent Statement:** Not applicable.

**Data Availability Statement:** The data that support the findings of this study are available from the corresponding author upon reasonable request.

**Conflicts of Interest:** The authors declare no conflict of interest.

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
