# Peer review of "Dissipativity Analysis of Large-Scale Networked Systems"

_applsci, doi:10.3390/app13021214_

Round 1
Reviewer 1 Report
The manuscript presents a study of dissipativity analysis for linear networked systems using a Linear Matrix Inequality (LMI) approach. The objective, as stated by the authors, is to reduce the computational burden of this type of analysis, which is demonstrated through numerical simulations. The approach is not new, but reduction in computational burden is an interesting problem.
If the manuscript is considered for publication, the authors should make the connection between the main theoretical results of the study (Section 3) to the reduction in computational cost observed in the calculation time (Section 4). As presented, it is not clear how the computational cost reduction is guaranteed to decrease, other than by allowing for more conservative dissipativity results, which is not necessarily a desired outcome in practice.
the key issue with the manuscript is that the computational cost reduction is not clearly linked with the main results of the study. The authors should probably fix this disconnection before publication.
Reviewer 2 Report
The Manuscript analyse dissipativity of large-scale networked systems with linear time-invariant dynamics. The authors have derved a condition for analysing the networked system including advantages of the block-diagonal structure of the system parameter matrix and the sparseness characteristics of the subsystem interconnections.
Overall the paper is well-written and organised. The contents are prepared to a high standard and the discussion is concise and informative. However, the authors should include a discussion on the relevance of this work for practical applications, at least with one example. Once included the manuscript can be accepted for publication.
